# Diagnostic Challenges and Risk Stratification of Periprosthetic Joint Infection in Patients with Inflammatory Arthritis

**DOI:** 10.3390/jcm14124302

**Published:** 2025-06-17

**Authors:** Paweł Kasprzak, Wiktoria Skała, Mariusz Gniadek, Adam Kobiernik, Łukasz Pulik, Paweł Łęgosz

**Affiliations:** 1Department of Orthopedics and Traumatology, Medical University of Warsaw, 4 W.H. Lindleya St, 02-005 Warsaw, Poland; pawel.legosz@wum.edu.pl; 2Military Institute of Medicine–National Research Institute, 128 Szaserów St, 04-141 Warsaw, Poland; mgniadek@wim.mil.pl; 3Samodzielny Publiczny Zespół Zakładów Opieki Zdrowotnej, 7 H. Sienkiewicza St, 09-100 Płońsk, Poland; wiktoria.skala99@gmail.com; 4Warszawski Szpital Południowy, 99 R. W. Pileckiego St, 02-798 Warsaw, Poland; adam.kobiernik@szpitalpoludniowy.pl

**Keywords:** arthroplasty, periprosthetic, infection, inflammatory arthritis, rheumatoid arthritis, infection markers

## Abstract

**Background/Objectives:** Accurate detection of periprosthetic joint infection (PJI) in patients with inflammatory arthritis (IA), including rheumatoid arthritis (RA), remains challenging due to overlapping inflammatory parameters and the influence of immunosuppressive regimens. **Methods:** A narrative review was conducted using PubMed/MEDLINE (2010–2025). Search terms included “periprosthetic joint infection”, “inflammatory arthritis”, “rheumatoid arthritis”, “diagnosis”, “biomarkers”, “synovial fluid”, and “immunosuppression”. Eventually, 50 studies were included. **Results:** IA patients diagnosed with PJI are more frequently younger, female, and present with a higher burden of comorbidities and an increased rate of false-positive histological findings and culture-negative infections. Standard biomarkers, such as serum C-reactive protein (CRP), erythrocyte sedimentation rate (ESR), as well as synovial fluid white blood cell count and polymorphonuclear leukocyte percentage, have a low to moderate value for diagnosing PJI in patients with IA. Optimal thresholds for these tests differ from those recommended by the Musculoskeletal Infection Society (MSIS). Alpha-defensin has demonstrated superior diagnostic performance among synovial fluid biomarkers included in MSIS criteria. Novel markers, such as serum bactericidal permeability-increasing protein (BPI) and neutrophil elastase-2 (ELA-2), as well as synovial C-reactive protein and calprotectin, along with molecular techniques like polymerase chain reaction (PCR), are showing increasing potential. **Conclusions:** Disease and treatment-related confounders hinder PJI diagnosis in IA. Adjusted thresholds and IA-specific approaches are needed. Further research should validate emerging biomarkers, among which BPI, ELA-2, and synovial CRP show the greatest diagnostic potential and guide perioperative immunosuppressive strategies.

## 1. Background

Total joint arthroplasty of major joints such as the hip (THA) and knee (TKA) remains the primary intervention for advanced osteoarthritis. It allows for optimal pain control and improvement in limb motor function [1]. However, like any surgical procedure, it carries the risk of postoperative complications, with one of the most serious being a periprosthetic joint infection (PJI). PJI can present as either an acute, rapidly progressing infection or a chronic, insidious process with persistent inflammation, leading to joint dysfunction. In severe cases, it can cause sepsis or death. Short-term mortality rates due to PJI are estimated to be between 5% and 10%. Furthermore, long-term mortality rates may be even higher, particularly in elderly patients with multiple comorbidities, such as diabetes, obesity, immunosuppressive therapy, autoimmune diseases, malnutrition, and smoking [2].

Diagnosis of PJI follows criteria established by the Musculoskeletal Infection Society (MSIS), with revisions introduced in 2018 by the International Consensus Meeting (ICM). According to these recommendations, infection can be confirmed if at least one of the two major criteria is met or at least six points are obtained from the minor criteria. Laboratory tests included in the classification are serum markers such as erythrocyte sedimentation rate (ESR), C-reactive protein (CRP) or D-dimer concentration, as well as synovial fluid parameters: white blood cell count (sWBC), polymorphonuclear percentage (PMN%), leukocyte esterase (LE), and alpha-defensin [3,4].

Although the updated PJI definition performed well in the general population (sensitivity and specificity reached 97.7% and 99.5%, respectively), its authors underline that the proposed criteria may lack precision in cases involving inflammatory arthritis (IA) exacerbations due to the confounding effects of systemic inflammation, which necessitates an individualized diagnostic approach in this patient group [5,6].

Rheumatoid arthritis (RA) is one of the most common types of inflammatory arthritis, with an estimated prevalence of approximately 1% in the general population [7]. The pathophysiology of RA involves immune dysregulation, resulting in excessive lymphocyte activation and pro-inflammatory cytokine release. Clinical symptoms include joint pain, swelling, morning stiffness, and restricted mobility, with a fluctuating course of exacerbations and remissions. The diagnosis of RA is based on inflammatory markers, such as CRP and ESR levels in the blood, along with the presence of rheumatoid factor (RF) in both blood and synovial fluid. Elevated levels of these markers indicate ongoing inflammation and correlate with disease severity. Serological tests, such as anti-cyclic citrullinated peptide antibody (anti-CCP) levels, further aid in diagnosis. Early stages of RA are also noticeable in the synovial fluid by the presence of inflammatory cells such as leukocytes and RF. RA treatment focuses on early diagnosis and disease-modifying therapy, including nonsteroidal anti-inflammatory drugs (NSAIDs), disease-modifying antirheumatic drugs (DMARDs), and biologic therapies [8].

The incidence of PJI among patients with RA undergoing total hip or knee replacement is up to 1.6 times greater than in patients undergoing the same procedures for osteoarthritis [9]. This is caused by the disease activity and its pharmacological treatment, which pertains especially to glucocorticoids (e.g., prednisone) and biologic DMARDs, such as anti-tumor necrosis factor (anti-TNF) inhibitors [5,10,11]. Treatment of infections in patients with persistent inflammation is difficult and requires a personalized, multidisciplinary approach.

The objectives of this narrative review are to (1) examine demographic patterns and predisposing factors for PJI in patients with inflammatory arthritis (2) analyze the challenges in applying the currently used criteria for PJI diagnosis in this patient group (3) propose directions for future research.

## 2. Methods

Source retrieval for the review was conducted between October 2024 and March 2025 by searching the MEDLINE/PubMed database using “periprosthetic joint infection” and “inflammatory arthritis”, which was followed by screening for keywords such as “diagnosis”, “differentiation”, “inflammatory markers”, and “disease flare”. Next, titles and abstracts of the obtained publications were assessed for their eligibility, study design, and relevance to the topic. Eligible studies included English-language publications from 2010 to 2025 involving human participants, using validated PJI definitions as reference standards and full-text availability: Musculoskeletal Infection Society (MSIS) 2011, MSIS modified by International Consensus Meeting (ICM) 2013 and 2018, Infectious Diseases Society of America (IDSA), European Bone and Joint Infection Society (EBJIS) 2021 5) evaluating PJI diagnosis after total hip arthroplasty (THA) and/or total knee arthroplasty (TKA). The search did not limit the type of reported inflammatory and non-inflammatory arthritis. In the end, 50 articles were enrolled for the review. Data from chosen sources were extracted and grouped into the following areas: (1) demography and risk factors of patients with inflammatory arthritis with PJI diagnosis, (2) efficacy and diagnostic value of preoperative and intraoperative tests for PJI diagnosis in patients with IA, both included and not included in the MSIS criteria.

## 3. Results

### 3.1. Assessment of Bias and Level of Evidence of the Enrolled Studies

The studies chosen for the extraction of data presented in this section in the majority are retrospective cohort studies and are, therefore, subject to selection bias. Although some enrolled large numbers of patients, the number of the IA PJI cohort remains small. Significant differences between the inflammatory and non-inflammatory arthritis (nIA) groups could influence the statistical analysis. Moreover, studies varied widely in both their IA and nIA cohorts due to inconsistent inclusion or exclusion of specific inflammatory diseases and discrepancies in how fractures and neoplasms were handled as aseptic control cases. The methodologies also lacked coherence, as evidenced by the varied inclusion of nIA groups as controls across studies. The authors also outlined that IA patients presented with various stages of inflammatory activity, DMARD usage, and a variety of comorbidities, which may have led to false-positive or false-negative results. For this reason, the level of evidence of these studies in the majority remains low to moderate. A comparison of the studies in terms of their methodology is presented in Table 1.

### 3.2. Demography Data and Risk Factors of Patients with Inflammatory Arthritis with PJI Diagnosis

Gender, age, and BMI. Compared to patients with noninflammatory arthritis (nIA), individuals with IA and PJI are predominantly female, of younger age, and tend to have lower body mass index values. Female patients represented 56.7–88.9% of the IA PJI cohort [12,13,14,15,16,17,18,19,20,21], with the age ranging from 48–78 years [12,13,14,15,17,18,19,20,21,22] and BMI 23–30.2 [12,13,15,19]. This tendency, however, is not uniquely determined. Only two studies stated a statistically significant difference in sex proportions and age between IA and nIA patients with PJI [13,21]. On the other hand, other authors found no significant differences in gender, age, and BMI between these two patient groups [18,20,22].

Reported types of inflammatory and noninflammatory arthritis diseases. In the vast majority of studies, the most commonly analyzed IA was rheumatoid arthritis, whereas the major or the only reported nIA was osteoarthritis. Other reported IAs were systemic lupus erythematosus (SLE), ankylosing spondylitis, gouty arthritis, psoriatic arthritis, Sjögren syndrome, Still disease, ulcerative colitis, polymyalgia rheumatica, sarcoidosis, polymyositis, systemic sclerosis, and juvenile RA [12,13,14,16,17,21,22]. Some authors excluded SLE and sarcoidosis as they were not considered IA [12,22]. In terms of nIA, one study also included posttraumatic arthritis, osteonecrosis, femoral neck fracture, developmental hip dysplasia, and neoplasms [21].

Comorbidities. Patients with IA who develop PJI often present with elevated Charlson Comorbidity Index scores in contrast to their nIA counterparts. In the study conducted by Shohat et al., 61.9% of IA PJI patients had a CCI score >=2, whereas in the nIA PJI group, this was present in 32.1% of the cohort [12]. Sculco et al. reported a significant difference in CCI between IA and nIA PJI groups, with an average of 2.8 for IA and 1.7 for nIA [13].

Pathogens. Staphylococcal species were identified as the predominant causative agents of PJI in both IA and nIA patient cohorts. In the IA PJI, staphylococcal species constituted 50–82% of all identified pathogens, but the differences in their occurrence between these two groups were not significant [13,18,21,22,23].

Rate of PJI. Whether patients with IA exhibit a higher incidence of PJI compared to those with nIA remains a matter of ongoing debate. Although Zhao et al. and Cipriano et al. claimed a significantly higher PJI rate in IA compared to nIA patients (6.92–31% vs. 0.67–18%) in a 5–7 year follow-up, Jiang et al. found no significant differences in the rate of PJI and superficial infections between IA and nIA groups [18,21,24]. Lai et al. reported significantly more infections in the IA group in general, including both PJI and superficial infections, but found no significant differences in the rate of PJI alone [25].

### 3.3. Efficacy and Diagnostic Value of Preoperative and Intraoperative Criteria for PJI Diagnosis in Patients with Inflammatory Arthritis

Preoperative standard MSIS markers. The MSIS minor diagnostic criteria for PJI encompass several preoperative indicators, including serum levels of C-reactive protein (CRP), D-dimer, and erythrocyte sedimentation rate (ESR), as well as synovial WBC, PMN%, alpha-defensin, and leukocyte esterase. Because the updated PJI definition was developed and validated on a patient group with chronic PJI, the review of its application for inflammatory arthritis pertains to chronic PJI only. Among the articles that measured their efficacy and diagnostic value for chronic PJI diagnosis in IA patients, PJI was diagnosed according to MSIS or EBJIS criteria. Major criteria were considered as apparent evidence of infection, even in the presence of chronic inflammation. The mean size of the IA patient cohort in the studies was relatively small, reaching 34 (range 8–60) [12,15,16,17,18,21,23]. For each preoperative marker, a receiver operating characteristic curve was established, and the optimal cut-off point was determined by the Youden Index. The Area Under the Curve (AUC) was calculated as a representative of the diagnostic value. Their efficacy was evaluated by the sensitivity, specificity, positive predictive value (PPV), and negative predictive value (NPV).

Studies showed that the optimal cut-off points recommended by the MSIS update for these markers may be incompatible for patients with inflammatory arthritis. The thresholds determined for each marker in the IA cohort differed between studies and lay within a wide range of values. For serum CRP and ESR, the cut-off points were at least equal to or higher than those recommended by the MSIS. In terms of synovial WBC and PMN%, some studies similarly calculated higher thresholds for IA patients; however, the tendency was not clear, as some determined them to be equal or lower than the MSIS values. The comparison between cut-off values for diagnostic tests proposed by the updated MSIS criteria and values determined for patients with inflammatory arthritis is presented in Table 2.

Determining the efficacy of most of these tests for patients with inflammatory arthritis remains unresolved, as the calculated values for sensitivity, specificity, PPV, and NPV differed significantly between studies, preventing the drawing of definitive conclusions. In terms of evaluating their diagnostic value, the majority of authors were consistent in claiming that they are better suited for patients without inflammatory arthritis. In the IA PJI cohort, authors showed a low-to-moderate diagnostic value for serum ESR, CRP, and D-dimer (AUC in the 0.6–0.9 range) and a moderate diagnostic value for synovial fluid WBC and PMN% (AUC 0.7–0.93) [12,15,16,17,18,21,23]. Among the evaluated markers, alpha-defensin demonstrated the highest diagnostic performance, with sensitivity reaching 92–93%, specificity 98–100%, and an AUC of 0.96–0.97 [16,17]. Additionally, a combined index of CRP, ESR, sWBC, and PMN% also achieved good efficacy and a high diagnostic value (sensitivity 80.5%, specificity 100%, AUC 0.944) [15].

The diagnostic application of leukocyte esterase (LE) in IA patients with suspected PJI has not been extensively studied, as only one study by Zhang et al. was found, which studied the influence of inflammatory arthritis on LE strip results. They found that the strip test could be influenced by IA when used for diagnosing PJI; among 20 samples with inflammatory arthritis without PJI, the LE strip of 18 samples was positive, among which 3 remained positive after centrifugation [26].

Intraoperative standard MSIS markers. The intraoperative markers found in minor ICM criteria are a single positive culture, positive histology, and positive purulence. None of the enrolled articles reported the rate of positive purulence findings in IA patients operated on for suspected PJI. Among the 3 studies that provided data on the culture results for IA patients operated on for suspected PJI, it was shown that among IA patients evaluated for suspected PJI, the proportion of culture-negative cases ranged between 16% and 30% [13,15,21]. Furthermore, Sculco et al. emphasized that a preoperative distinction of culture-negative patients was difficult. Between culture-positive IA PJI and culture-negative IA PJI, no significant differences were found in the demography, such as age, gender, comorbidities, use of DMARDs, as well as in laboratory tests, including ESR, CRP, sWBC, and PMN% [13].

In terms of histology testing, studies showed that chronic inflammation in IA patients may hinder its reliable interpretation in terms of PJI diagnosis. Shohat et al. reported that patients with aseptic IA were approximately twice as likely to yield false-positive histological findings compared to those with noninflammatory conditions [12]. On the other hand, Sculco et al. underlined that IA patients with culture-positive PJI had a significantly larger proportion of positive histology for acute infection than IA patients with culture-negative PJI [13].

Non-MSIS biomarkers. Apart from the markers involved in the MSIS criteria, some studies were found to investigate less common markers for PJI diagnosis in patients with inflammatory arthritis. These included serum human neutrophil elastase-2 (ELA-2), bactericidal permeability-increasing protein (BPI), and procalcitonin, as well as synovial fluid CRP, monocyte cell count, PMN cell count, and percentage of monocytes [17,18,23]. Among them, serum ELA-2, BPI, and procalcitonin, as well as synovial fluid CRP, achieved a high diagnostic value (AUC 0.9–0.95), suggesting that they may help aid PJI diagnosis for IA patients. The measured values of the sensitivity, specificity, PPV, NPV, and AUC for each reviewed marker for PJI diagnosis in IA patients are presented in Table 3. A visual summary of biomarkers’ performance is presented in a bubble plot in Figure 1.

Promising biomarkers requiring further research on an inflammatory arthritis cohort. Among the markers that have not yet been broadly investigated in patients with inflammatory arthritis for PJI diagnosis, calprotectin in synovial fluid has shown considerable potential as a diagnostic marker for PJI. It is a protein found in neutrophils, playing a crucial role in antimicrobial defense. Studies are consistent in reporting its superiority in PJI diagnosis compared to other synovial markers, such as WBC, LE, and CRP, and comparable performance to alpha-defensin [27]. Meta-analyses revealed that for PJI diagnosis, calprotectin achieved a pooled sensitivity and specificity of up to 94% and 93%, respectively [28,29]. However, the matter of implementing the biomarker for standard clinical practice for PJI diagnosis in IA patients requires further research. Calprotectin levels have been proven to be correlated with IA activity, and it is not yet concluded whether IA patients require a higher calprotectin threshold for PJI diagnosis [30,31].

Molecular diagnostics. Furthermore, diagnosing PJI in culture-negative IA patients may also be aided by molecular techniques, which are increasingly recognized for their potential role in such cases. Culture-negative PJIs commonly result from prior antibiotic use, infection with slow-growing organisms, or pathogen-associated protective mechanisms such as biofilm formation. In such scenarios, molecular diagnostics like polymerase chain reaction (PCR) or next-generation sequencing (NGS) may provide substantial diagnostic value. When clinical suspicion of infection remains high despite negative cultures, targeted PCR assays for specific pathogens or commercially available multiplex PCR kits that detect common musculoskeletal infection agents may be employed. A novel strategy involves broad-range PCR targeting the 16S ribosomal RNA gene, which is conserved across all bacterial species, followed by sequencing via Sanger or NGS methods. NGS represents a highly sensitive and specific alternative that does not require predefined primers, as PCR does. However, although PCR was shown to be valuable for ruling out PJI due to its relatively high specificity of 0.86, which is similar to other diagnostic tools such as serum CRP, synovial fluid WBC counts, and synovial fluid WBC differentiation, its widespread application is limited by high costs and availability in comparison to those methods. Another significant limitation of PCR is the inaccessibility of specific probes for common low-virulence organisms (e.g., *S. epidermidis* or *C. acnes*) in some widely used assays. NGS, on the other hand, requires access to a comprehensive bacterial genome database and is susceptible to sample contamination tests, which occasionally detect microorganisms not previously reported in human infection, a result that some authors attribute to a “native microbiome” [32,33,34]. Moreover, samples are susceptible to contamination during collection or laboratory analysis. These factors may lead to false-positive results. The high cost of this method also limits its use; since some studies have demonstrated that NGS sensitivity is comparable to that of conventional culture, its application should be restricted to culture-negative IA [32,35,36,37].

Considering the performance of the markers mentioned above, we would like to propose a suggestion for an algorithm for PJI diagnosis in patients with inflammatory arthritis (Figure 2). We wish to disclose, however, that this would require thorough clinical validation before being introduced in practice. This validation process involves the systematic establishment of optimal cut-off points and the assessment of currently available tests’ effectiveness, alongside the evaluation of novel techniques, specifically for the IA patient cohort.

## 4. Discussion

Although efforts have been made to refine cut-off values for standard laboratory indicators in IA-related PJI, a universally accepted consensus is yet to be reached. As a result, diagnosis as well as treatment of PJI in patients with inflammatory arthritis is complicated and necessitates an individualized, multidisciplinary approach. These difficulties are closely related to several risk factors resulting from the disease activity and the pharmacological management of chronic inflammation.

### 4.1. Diagnostic Challenges

Critical Appraisal of Current PJI Definitions. Given the diagnostic uncertainty in this population, some authors have questioned whether the standard MSIS/ICM definitions for PJI are sufficiently sensitive or specific in patients with active inflammatory arthritis [38]. The elevated baseline of inflammatory markers in IA patients may lead to over-diagnosis of PJI, while immunosuppressed states may mask typical responses to infection. As such, it has been proposed that adjusted diagnostic criteria or alternate marker thresholds should be considered for this subgroup. Although no consensus exists yet, preliminary work by Goodman et al. [39] and Wouthuyzen-Bakker et al. [40] suggests the need for stratified diagnostic algorithms to account for disease activity and immunosuppressive therapy.

Although the systemic activity of IA may raise serum markers even in aseptic revisions, some studies suggest that reliable differentiation between septic and aseptic cases is still possible. Comparative analyses indicate that inflammatory markers such as ESR, CRP (serum and synovial), synovial WBC count, PMN%, and procalcitonin tend to be significantly higher in IA patients with confirmed PJI than in those undergoing aseptic revision [41].

Only a few studies have directly addressed the diagnostic challenge of distinguishing PJI from IA flares. Some evidence suggests that synovial fluid analysis—including PMN% and total leukocyte count, combined with newer biomarkers such as alpha-defensin may improve specificity. However, false positives can still occur in patients with active systemic inflammation, particularly during flares [37,42,43].

Moreover, immunosuppressive treatment—including glucocorticoids and JAK inhibitors—can further complicate the interpretation of laboratory results. Chronic exposure to these agents has been associated with dose-dependent persistently elevated CRP in the absence of infection, as well as an attenuated inflammatory response during actual infection. These effects may obscure the typical diagnostic profile of PJI and highlight the need for interpretation within the full clinical context, including disease activity and treatment history [44].

### 4.2. Perioperative Risk Factors and Medication Management

The risk of infection in IA patients is multifactorial, influenced by both disease activity and treatment strategies. Several observational studies [25,45,46,47,48,49] have shown that higher disease activity is associated with an increased risk of infection and poor outcomes after arthroplasty. Chronic inflammation may also contribute to anemia, malnutrition, or impaired immunity, which compromise healing.

Glucocorticoids and biologic DMARDs represent key pharmacologic risk factors. Salt et al. [45] and Wang et al. [15] reported that patients exposed to glucocorticoids, especially in the perioperative window, have significantly higher rates of infection. Similarly, patients continuing biologic therapy through surgery face a 3- to 5-fold higher risk of PJI. Momohara et al. [20] noted this elevated risk even when biologics were properly withheld, suggesting possible long-term immunosuppressive effects.

With the advent of newer agents such as Janus kinase (JAK) inhibitors (tofacitinib, baricitinib, upadacitinib), concerns have been raised regarding both thrombotic and infectious complications. Although perioperative data are still limited, observational studies suggest that prolonged exposure and higher cumulative doses of JAK inhibitors may elevate the risk of postoperative infections. These findings highlight the need to consider both the class of immunosuppressive therapy and the duration of exposure when planning surgery [46,47,49].

### 4.3. Guidelines and Clinical Implications

The 2022 ACR/AAHKS guideline revisions advocate for a personalized, risk-adjusted strategy. Recommendations include: (a) withholding biologic agents according to dosing cycles, (b) avoiding perioperative glucocorticoid exposure unless clinically essential, and (c) coordinated care between rheumatologists, orthopedic surgeons, and patients. Importantly, challenges in diagnosis and perioperative management should be viewed as separate but interlinked issues. Diagnostic complexity stems from altered inflammatory responses and the effects of immunosuppressive therapy, whereas perioperative infection risk is a function of pharmacologic exposure and disease severity [39,50].

### 4.4. Limitations

Our review has several limitations. It is a narrative review; no systematic analysis or meta-analysis was performed on the efficacy and diagnostic value of PJI markers in IA patients. The proposed diagnostic algorithm requires future validation. Additionally, reviewed studies are based on small, heterogeneous populations and varying methodologies. Lastly, this review focuses specifically on chronic periprosthetic joint infection (PJI), as most of the available literature addresses chronic cases. We deliberately limited the scope to allow for a more precise evaluation of diagnostic criteria and biomarker performance in this context. Including acute PJI would have introduced significant heterogeneity. However, acute PJI remains clinically relevant and deserves dedicated investigation in future studies.

## 5. Conclusions

### 5.1. Risk Factors for PJI in Patients with Inflammatory Arthritis

Patients with IA undergoing alloplasty have an increased risk of infection due to disease activity and IA treatment.Key contributors to increased PJI risk include elevated systemic inflammation, extended disease duration, corticosteroid use, and the uninterrupted administration of biologic agents during the perioperative phase.Future studies should determine a more accurate optimal timing of withholding IA therapy and consider the role of comorbidities such as metabolic syndrome or cardiovascular disease in the pathogenesis of PJI in patients with inflammatory arthritis.

### 5.2. Suggestions for PJI Diagnosis in Patients with Inflammatory Arthritis

In patients with inflammatory arthritis, diagnostic efforts should primarily rely on fulfilling major MSIS criteria, such as dual positive cultures or the presence of a sinus tract.The application of the scoring system recommended in the updated MSIS criteria for cases that do not meet the main criteria is limited in patients with autoimmune inflammation due to the low efficacy of the available diagnostic tests. Adhering to recommended cut-off points may lead to false-positive interpretations.Synovial fluid markers such as sWBC, PMN%, and alpha-defensin are relatively the most reliable and may be most helpful in diagnosing uncertain infection cases. Serum ESR and CRP can be applied in combination with synovial markers.Additional markers not included in the MSIS criteria, such as serum ELA–2, BPI, procalcitonin, synovial CRP, calprotectin, and molecular techniques like PCR present promising diagnostic values for the diagnosis of PJI, but more studies are needed to confirm their efficacy for patients with IA.To advance diagnostic accuracy in IA-related PJI, additional clinical studies are necessary to formulate and validate a tailored diagnostic algorithm or scoring model.

## Figures and Tables

**Figure 1 jcm-14-04302-f001:**
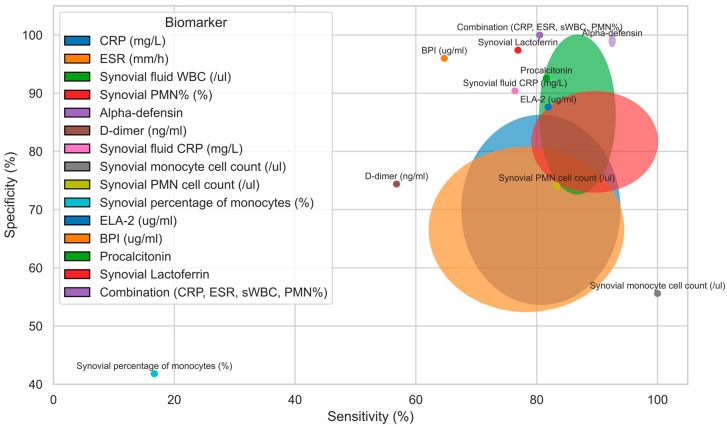
A bubble plot of the sensitivity and specificity of laboratory markers in diagnosing chronic PJI in patients with inflammatory arthritis.

**Figure 2 jcm-14-04302-f002:**
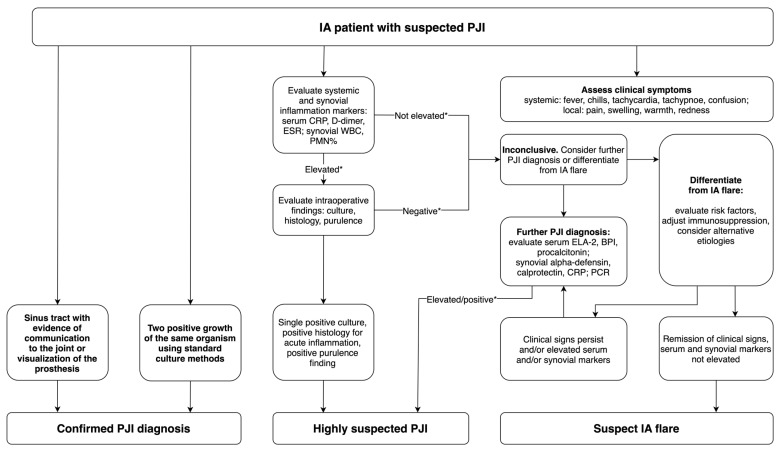
Proposed suggestion of an algorithm for PJI diagnosis in patients with inflammatory arthritis. * Cut-off points recommended by MSIS may not be applicable for patients with inflammatory arthritis. Positive histology-more than 5 neutrophils per high-power field in 5 high-power fields observed from histological analysis of periprosthetic tissue at 400× magnification.

**Table 1 jcm-14-04302-t001:** Comparison of chosen studies in terms of their methodology.

Study	Study Type	Sample Size of IA PJI	Sample Size of nIA PJI	Sample Size of IA nPJI	Sample Size of nIA nPJI	Type of PJI	Definition of an Acute PJI
Shohat et al. [12]	Retrospective	55	512	61	592	Chronic	Less than 3 months from surgery
Sculco et al. [13]	Retrospective	36	771	N/A	N/A	Not specified	N/A
Xu et al. [14]	Retrospective	30	N/A	32	N/A	Acute and chronic *	Less than 4–6 weeks from surgery
Wang et al. [15]	Retrospective	60	104	80	104	Chronic	Less than 3 months from surgery
Miyamae et al. [16]	Retrospective	41	N/A	N/A	N/A	Chronic	Less than 4–6 weeks from surgery
Tahta et al. [17]	Prospective	17	N/A	21	N/A	Not specified	N/A
Zhao et al. [18]	Retrospective	40	102	538	15,022	Chronic	Less than 3 months from surgery
De Araujo et al. [19]	Retrospective	53	N/A	N/A	N/A	Acute and chronic	Definition not provided
Momohara et al. [20]	Retrospective	3	N/A	417	N/A	Not specified	N/A
Cipriano et al. [21]	Prospective	19	146	42	664	Chronic	Less than 3 months from surgery
Carlson et al., [22]	Retrospective	26	N/A	58	N/A	Not specified	N/A
Ren et al. [23]	Retrospective	17	121	N/A	N/A	Acute and chronic	Less than 4–6 weeks from surgery
Jiang et al. [24]	Retrospective	1	1	219	260	Not specified	N/A
Lai et al. [25]	Retrospective	4	1	333	336	Not specified	N/A

* Authors did not analyze acute and chronic PJI cohorts separately.

**Table 2 jcm-14-04302-t002:** Comparison between cut-off values for diagnostic tests proposed by the updated MSIS criteria and cut-off values determined for patients with inflammatory arthritis.

	Predictive Cutoff (IA)	Predictive Cutoff (MSIS) ⊕
CRP (mg/L)	10.0–29.05 ∗	10.00
D-dimer (ng/mL)	796.50 •	860
ESR (mm/h)	30–39 °	30
Synovial fluid WBC (/ul)	1948–3654 +	3000
Synovial PMN% (%)	65.9–85.3 #	70
Alpha-defensin	Positive ⊥	Positive

CRP, C-reactive protein; ESR, erythrocyte sedimentation rate; WBC, white blood cells; PMN, polymorphonuclear cells. Note, predictive cut-off points determined for each test come from the following sources: ∗ ([12,15,16,17,18,21,23]), • ([18]), ° ([12,15,16,17,18,21]), + ([12,15,16,17,21,23]), # ([12,15,17,21,23]), ⊥ ([16,17]), ⊕ ([4]).

**Table 3 jcm-14-04302-t003:** Accuracy of laboratory markers in diagnosing chronic PJI in patients with inflammatory arthritis.

	Authors	Predictive Cutoff	Sensitivity (%)	Specificity (%)	Positive Predictive Value (%)	Negative Predictive Value (%)	AUC
CRP (mg/L)	[12,15,16,17,18,21,23]	10.0–29.05	67.6–93.8	53.7–86.2	18.8–86.2	73.4–96.0	0.676–0.920
ESR (mm/h)	[12,15,16,17,18,21]	30–39	62.2–94.4	52.5–80.7	53.0–85.8	66.4–96.0	0.613–0.890
Synovial fluid WBC (/ul)	[12,15,16,17,21,23]	1948–3654	80.5–93.0	72.7–100.0	25.0–100.0	68.0–97.6	0.780–0.938
Synovial PMN% (%)	[12,15,17,21,23]	65.9–85.3	79.2–100.0	73.0–90.3	18.5–89.7	70.0–100.0	0.710–0.936
Alpha- defensin	[16,17]	P	92.0–93.0	98.0–100.0	100	96	0.960–0.970
D-dimer (ng/mL)	[18]	796.50	56.8	74.4	79.8	66.4	0.657
Synovial fluid CRP (mg/L)	[17]	11.7	76.4	90.4	ND	ND	0.920
Synovial monocyte cell count (/ul)	[23]	830	100	55.6	20.0	100.0	0.750
Synovial PMN cell count (/ul)	[23]	1618	83.3	74.1	26.3	97.6	0.800
Synovial percentage of monocytes (%)	[23]	14.7	16.7	41.8	3.00	82.1	0.69
ELA-2 (ug/mL)	[17]	1.9	81.9	87.6	ND	ND	0.950
BPI (ug/mL)	[17]	3.47	64.7	96.0	ND	ND	0.920
Procalcitonin	[17]	0.1	81.6	92.6	ND	ND	0.930
Synovial Lactoferrin	[17]	9.1	76.9	97.4	ND	ND	0.900
Combination (CRP, ESR, sWBC, PMN%)	[15]	N/A	80.5	100.0	100.0	69.2	0.944

AUC, area under the curve; ELA-2, human neutrophil elastase 2; BPI, bactericidal permeability-increasing protein.

## Data Availability

No new data were created or analyzed in this study.

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
