# Peer review of "Diagnostic Challenges and Risk Stratification of Periprosthetic Joint Infection in Patients with Inflammatory Arthritis"

_jcm, 2025, doi:10.3390/jcm14124302_

Round 1

Reviewer 1 Report

Comments and Suggestions for Authors

This is a narrative review on PJI in patients with IA, focusing on the diagnostic challenges and risk stratification of PJI in IA patients (such as rheumatoid arthritis). The study reviewed literature from MEDLINE/PubMed database (2010–2025) to analyze the clinical characteristics, diagnostic marker efficacy, and management strategies for PJI in IA patients. Paweł Kasprzak et al. present a clear logical framework with coherent sentences and a comprehensive description. However, there are still areas that require improvement. Here are my suggestions:

(1) This is a narrative review and does not employ the standardized methods of systematic review or meta-analyse. Literature screening and data extraction may be subject to subjective bias, and there is a lack of quantitative assessment of the level of evidence;

(2) The included studies have small sample sizes and high heterogeneity (e.g., different types of IA, disease severity and activity, comorbidities, and immunosuppressive treatment regimens (specific use of different biologics)), leading to insufficient comparability of results. Although the authors acknowledged these limitations in the discussion of limitations;

(3) The analysis of the association between immunosuppressive therapy (e.g., JAK inhibitors) and PJI risk is insufficient, and the impact of drug dosage and duration of use on diagnostic markers has not been thoroughly explored;

(4) The article has a duplication rate of 22%. If possible, please reduce the duplication rate.

Author Response

Thank you very much for taking the time to review this manuscript. Please find the detailed responses below. The corresponding revisions/corrections are highlighted in red in the resubmitted file.

Thank you for providing a detailed scoring of our manuscript. We interpret this feedback as affirming the overall strength of our work while indicating the need for significant improvement in our referencing.

Comment 1: This is a narrative review and does not employ the standardized methods of systematic review or meta-analyse. Literature screening and data extraction may be subject to subjective bias, and there is a lack of quantitative assessment of the level of evidence.

Response 1: Thank you for your valuable feedback. We agree with your comment regarding the nature of the review and the need for a clearer assessment of the evidence.

Therefore, we've updated the manuscript to include a dedicated section that thoroughly discusses the potential biases and evaluates the level of evidence for the enrolled studies. This new section "3.1. Assessment of bias and level of evidence of the enrolled studies”, can be found on page 3, starting at line 112. In this section, we acknowledge that the majority of included studies are retrospective cohort designs, which are subject to selection bias and other limitations. We also address the heterogeneity and inconsistencies in methodology across studies, leading to our conclusion that the overall level of evidence remains low to moderate. A detailed comparison of the studies' methodologies is now provided in a new Table (Table 1).

We believe these additions significantly enhance the transparency of our review, addressing the concerns raised.

Comment 2: The included studies have small sample sizes and high heterogeneity (e.g., different types of IA, disease severity and activity, comorbidities, and immunosuppressive treatment regimens (specific use of different biologics)), leading to insufficient comparability of results. Although the authors acknowledged these limitations in the discussion of limitations.

Response 2: Thank you for your second comment. We entirely agree that the small sample sizes and high heterogeneity across the included studies are significant limitations affecting the comparability of results.

As we noted in our previous response, we've now explicitly addressed these very points in our revised manuscript. This acknowledgment and detailed discussion of the specific heterogeneities, which directly lead to the "low to moderate" level of evidence we discuss, can also be found within the “3.1. Assessment of bias and level of evidence of the enrolled studies "section.

Comment 3: The analysis of the association between immunosuppressive therapy (e.g., JAK inhibitors) and PJI risk is insufficient, and the impact of drug dosage and duration of use on diagnostic markers has not been thoroughly explored.

Response 3: Thank you for your feedback regarding the analysis of immunosuppressive therapy and its impact on PJI risk and diagnostic markers. We agree this area warrants a more thorough discussion.

We've accordingly revised the manuscript to address this point. You can find this updated section in the Discussion on page 11, starting at line 369. This revision now elaborates on how the duration and dosage of immunosuppressive drugs, including JAK inhibitors, can influence both infection risk and the interpretation of diagnostic markers, highlighting the need to consider these factors during PJI diagnosis in the IA patient group.

Comment 4: The article has a duplication rate of 22%. If possible, please reduce the duplication rate.

Response 4: Thank you for bringing the duplication rate to our attention. We understand the importance of minimizing redundancy and agree with your suggestion.

We have thoroughly reviewed the manuscript and have carefully paraphrased sections where we identified opportunities to reduce duplication. Our goal was to improve the flow and conciseness of the text while retaining all necessary information. We hope these revisions have successfully lowered the overall duplication rate. All changes can be seen in red color throughout the manuscript.

Once again, thank you for your thorough review which to a great extent helped us to increase the quality of our manuscript.

Kind regards,

Authors

Reviewer 2 Report

Comments and Suggestions for Authors

This narrative review comprehensively addresses the diagnostic challenges and risk stratification of periprosthetic joint infection (PJI) in patients with inflammatory arthritis (IA). It clearly outlines the complexities of diagnosing PJI in this population, given the confounding effects of inflammatory markers and immunosuppressive therapies. The article evaluates the limitations of current diagnostic criteria (e.g., MSIS criteria) and explores emerging biomarkers and diagnostic strategies. However, limitations such as methodological inconsistencies and data heterogeneity constrain the generalizability of some conclusions. revisions are needed to enhance its scientific rigor and clinical applicability.

As a narrative review rather than a systematic review or meta-analysis, the article’s conclusions lack the rigor of systematic evidence synthesis. The absence of bias assessment or data pooling may make some findings appear subjective.

The proposed algorithm in Figure 1 is promising but lacks clinical validation. The article acknowledges this but does not provide a clear plan for validation.

The review highlights the potential of molecular techniques (e.g., PCR, NGS) for culture-negative PJI but provides limited discussion of their limitations (e.g., high costs, sample contamination risks) and performance in IA patients specifically.

The review focuses solely on chronic PJI, excluding acute PJI, which limits its comprehensiveness since acute PJI is also clinically significant.

The discussion section is somewhat redundant, particularly regarding glucocorticoid and biologic risks, which are repeated across multiple paragraphs.

Author Response

Thank you very much for your thorough review and suggestions. Please find the detailed responses below. The corresponding revisions/corrections are highlighted in the resubmitted files in red color.

We interpret the rating as a constructive assessment highlighting our work's foundational strength and clear opportunities for further enhancement.

Comment 1: As a narrative review rather than a systematic review or meta-analysis, the article’s conclusions lack the rigor of systematic evidence synthesis. The absence of bias assessment or data pooling may make some findings appear subjective.

Response 1: Thank you for your feedback regarding the perceived subjectivity of some findings. We agree that the rigor of systematic evidence synthesis is paramount and appreciate you highlighting the importance of bias assessment.

We have, accordingly, revised the manuscript to address this point comprehensively. We've included a dedicated section, "3.1. Assessment of bias and level of evidence of the enrolled studies," which can be found on page 3, starting at line 112.

In this new section, we transparently discuss the potential biases inherent in the included studies, acknowledging that most are retrospective cohort designs and are thus subject to selection bias and other limitations. We also address the heterogeneity and methodological inconsistencies across these studies, which collectively lead us to conclude that the overall level of evidence remains low to moderate. To further enhance transparency, a detailed comparison of the studies' methodologies is now provided in a new Table (Table 1).

We believe these additions significantly enhance the scientific rigor of our narrative review by openly addressing its limitations and providing a structured assessment of the evidence base.

Comment 2: The proposed algorithm in Figure 1 is promising but lacks clinical validation. The article acknowledges this but does not provide a clear plan for validation.

Response 2: Thank you for your comment on the proposed algorithm in Figure 1 (now Figure 2) and the need for a clear validation plan. We agree that clinical validation is crucial for its practical application.

We have, accordingly, revised the manuscript to address this point more comprehensively. This updated section, including a clearer plan for future validation, can be found in the Results section on page 10, starting at line 329.

In this revision, we now explicitly state our commitment to the systematic establishment of optimal cut-off points and the assessment of existing tests' effectiveness, alongside the evaluation of novel techniques, all specifically within the IA patient cohort. We emphasize that this thorough clinical validation is a necessary next step for introducing the algorithm into practice.

Comment 3: The review highlights the potential of molecular techniques (e.g., PCR, NGS) for culture-negative PJI but provides limited discussion of their limitations (e.g., high costs, sample contamination risks) and performance in IA patients specifically.

Response 3: Thank you for this comment. We have included a more thorough description of limitations of mentioned methods. Also, we have compared the efficacy of PCR with other methods and explained why cost-effectiveness, beside technical limitations, is the main reason why the use of NGS should be considered only in culture-negative IA, not IA in general. This section can be found in the Results section on page 9, starting at line 313.

Comment 4: The review focuses solely on chronic PJI, excluding acute PJI, which limits its comprehensiveness since acute PJI is also clinically significant.

Response 4: Thank you for your insightful comment regarding the scope of our review. We've revised the manuscript to clearly state our deliberate focus on chronic periprosthetic joint infection, explaining that this allowed for a more precise evaluation of diagnostic criteria given that most of the available literature addresses chronic cases, thus avoiding the heterogeneity that including acute cases would introduce. This change can be found in the Limitations section of the Discussion, on page 12, starting at line 406.

Comment 5: The discussion section is somewhat redundant, particularly regarding glucocorticoid and biologic risks, which are repeated across multiple paragraphs.

Response 5: Thank you for your feedback concerning the redundancy in the discussion section. We agree it's crucial for the discussion to be concise and impactful.

We've accordingly revised the Discussion section to enhance its flow and eliminate repetitions, particularly concerning glucocorticoid and biologic risks. Due to major changes made, the whole Discussion section is marked in red. We've streamlined the information to present a clearer and more coherent narrative without losing any essential points.

Once again, thank you for your thorough review which to a great extent helped us to increase the quality of our manuscript.

Kind regards,

Authors

Reviewer 3 Report

Comments and Suggestions for Authors

This narrative review is highly topical and clinically relevant, addressing the nuanced and often overlooked diagnostic complexities of periprosthetic joint infection (PJI) in patients with inflammatory arthritis (IA). The authors provide a comprehensive synthesis of current evidence, incorporating updated diagnostic criteria, biomarker performance, and treatment-related risk factors. The manuscript is well-researched and clearly written.

The title accurately reflects the content and focus of the review.

The abstract is comprehensive and well-structured, summarizing the objectives, methods, key findings, and conclusions. In the abstract, briefly indicate the total number of included studies (n=37). Specify which markers or methods (e.g., alpha-defensin, calprotectin, PCR) show the greatest diagnostic promise, to highlight practical takeaways.

Consider explicitly stating the objective or research questions in the last paragraph of introduction to guide the reader.

Clarify how data from heterogeneous studies were managed or synthesized, given the narrative nature of the review. Indicate whether risk of bias or study quality was considered when interpreting results.

Consider organizing the results into clearer subheadings to improve readability (e.g., “Standard MSIS Markers,” “Non-MSIS Biomarkers,” “Molecular Diagnostics”).

The inclusion of a visual summary of key biomarker performance (e.g., a forest plot or bubble plot) could enhance understanding.

Discussion could be strengthened by more clearly separating challenges in diagnosis vs. those related to perioperative management.

Consider a brief critical appraisal of whether current PJI definitions should be modified for IA patients.

Author Response

We are grateful for your detailed review and valuable suggestions. Please find the detailed responses below. The corresponding revisions/corrections are highlighted in the resubmitted files in red color.

We appreciate that our organization and referencing were affirmed as strengths, while our contribution and scientific soundness were noted as areas for further refinement.

Comment 1: The abstract is comprehensive and well-structured, summarizing the objectives, methods, key findings, and conclusions. In the abstract, briefly indicate the total number of included studies (n=37). Specify which markers or methods (e.g., alpha-defensin, calprotectin, PCR) show the greatest diagnostic promise, to highlight practical takeaways.

Response 1: Thank you for pointing this out. We agree with this comment. Therefore, we have mentioned a number of included studies in the “Methods” section of the abstract (page 3, line 106). Please note that due to other revisions made, the number of references has changed. As for alpha-defensin, we have underlined it is the best diagnostic option in current MSIS criteria. With regard to novel biomarkers and diagnostic methods, we have chosen BPI, ELA-2 and synovial CRP as the most promising tools. The abbreviations of new biomarkers have been provided in the brackets.

Comment 2: Consider explicitly stating the objective or research questions in the last paragraph of introduction to guide the reader.

Response 2: Thank you for your helpful suggestion to explicitly state our objectives. We agree that this will greatly benefit the reader. We have, accordingly, revised the last paragraph of the Introduction to clearly outline the objectives of this narrative review. This change can be found on page 2, starting at line 85. In this updated section, we now explicitly state our aims to explore demographic patterns and predisposing factors for PJI in inflammatory arthritis patients, analyze the challenges in applying current diagnostic criteria for this group, and propose directions for future research.

Comment 3: Clarify how data from heterogeneous studies were managed or synthesized, given the narrative nature of the review. Indicate whether risk of bias or study quality was considered when interpreting results.

Response 3: Thank you for your question regarding the management of heterogeneous data and the consideration of study quality in our narrative review. We agree on the importance of transparently addressing these aspects.

We have revised the manuscript to explicitly clarify how we approached the interpretation of results from diverse studies. This clarification, including how we considered risk of bias and study quality, can be found in the new section 3.1, titled "Assessment of bias and level of evidence of the enrolled studies” located on page 3, starting at line 112.

In this new section, we detail that while most included studies are retrospective cohorts prone to selection bias and other limitations, we've carefully considered the methodological inconsistencies across studies, such as varying inclusion/exclusion criteria for inflammatory diseases and control groups. We also acknowledge the broad heterogeneity within patient cohorts regarding disease activity, DMARD usage, and comorbidities, which influenced our assessment that the overall level of evidence of these studies remains low to moderate. A comprehensive comparison of methodologies is also provided in a new Table (Table 1) to support our interpretations.

Comment 4: Consider organizing the results into clearer subheadings to improve readability (e.g., “Standard MSIS Markers,” “Non-MSIS Biomarkers,” “Molecular Diagnostics”).

Response 4: Thank you for your suggestion to improve the readability of our results section. We agree that clearer subheadings will greatly enhance the reader's experience.

We have, accordingly, revised the Results section by introducing new subheadings to categorize the findings more effectively. We've now organized the results under distinct headings such as "Preoperative standard MSIS markers," "Intraoperative standard MSIS markers," "Non-MSIS biomarkers," "Promising biomarkers requiring further research on an inflammatory arthritis cohort," and "Molecular diagnostics." We believe this refined structure makes it much easier to navigate and understand our findings.

Comment 5: The inclusion of a visual summary of key biomarker performance (e.g., a forest plot or bubble plot) could enhance understanding.

Response 5: Thank you for your excellent suggestion to enhance the visual representation of our findings. We agree that a visual summary of biomarker performance would greatly improve understanding.

We have, accordingly, modified the manuscript by including a bubble plot of the sensitivity and specificity of laboratory markers in diagnosing chronic PJI in patients with inflammatory arthritis. This new visual summary can be found as Figure 1 on page 9. We believe this addition provides a clear and concise overview of the key biomarker performance, making our results more accessible and impactful for the reader.

Comment 6: Discussion could be strengthened by more clearly separating challenges in diagnosis vs. those related to perioperative management.

Response 6: Thank you for your constructive feedback regarding the organization of our discussion section. We agree that a clearer separation between diagnostic challenges and perioperative management issues will enhance understanding.

We've accordingly revised the discussion to create a more comprehensible distinction between these two crucial areas. We believe this refined structure makes it easier for readers to grasp the specific challenges associated with each aspect of PJI in the context of inflammatory arthritis.

Comment 7: Consider a brief critical appraisal of whether current PJI definitions should be modified for IA patients.

Response 7: Thank you for suggesting we critically appraise whether current PJI definitions should be modified for inflammatory arthritis (IA) patients. We agree that this is a vital point for discussion.

We have, accordingly, revised the manuscript to include a critical appraisal of current PJI definitions in the context of IA patients. This addition can be found on page 11, starting at line 348. In this new section, we explore the challenges posed by diagnostic uncertainty in this population, discussing how factors like elevated baseline inflammatory markers and immunosuppression can affect the accuracy of standard PJI criteria. We also highlight proposed solutions, such as adjusted diagnostic criteria or stratified algorithms, acknowledging the need for further research in this area. We believe this addition significantly strengthens the depth and relevance of our discussion.

We put in the effort to improve our English to more clearly express the research. Once again, thank you for your thorough review which to a great extent helped us to increase the quality of our manuscript.

Kind regards,

Authors

Round 2

Reviewer 1 Report

Comments and Suggestions for Authors

The authors have made good revisions to the relevant content based on the reviewers' comments. The revised article is more comprehensive and coherent. We recommend accepting it in its current form.

Reviewer 3 Report

Comments and Suggestions for Authors

The Authors made good efforts in the attempt to ameliorate their paper.